# Elevated FAI Index of Pericoronary Inflammation on Coronary CT Identifies Increased Risk of Coronary Plaque Vulnerability after COVID-19 Infection

**DOI:** 10.3390/ijms24087398

**Published:** 2023-04-17

**Authors:** Botond Barna Mátyás, Imre Benedek, Emanuel Blîndu, Renáta Gerculy, Aurelian Roșca, Nóra Rat, István Kovács, Diana Opincariu, Zsolt Parajkó, Evelin Szabó, Bianka Benedek, Theodora Benedek

**Affiliations:** 1Clinic of Cardiology, Mureș County Emergency Clinical Hospital, 540142 Târgu Mureș, Romania; matyas_botond@yahoo.com (B.B.M.); imre.benedek@umfst.ro (I.B.); emi.blindu@yahoo.com (E.B.); gerculy_renata@yahoo.com (R.G.); rosca_aurelian@yahoo.com (A.R.); istvan.kovacs@umfst.ro (I.K.); p_zsolt92@yahoo.com (Z.P.); szaboevelin22@yahoo.com (E.S.); theodora.benedek@umfst.ro (T.B.); 2Center of Advanced Research in Multimodality Cardiac Imaging, CardioMed Medical Center, 540124 Târgu Mureș, Romania; diana.opincariu@yahoo.ro; 3Doctoral School of Medicine and Pharmacy, University of Medicine, Pharmacy, Science and Technology “George Emil Palade” of Târgu-Mures, 540139 Târgu-Mures, Romania; 4Department of Cardiology, University of Medicine, Pharmacy, Science and Technology “George Emil Palade” of Târgu-Mures, 540139 Târgu Mureș, Romania; 5Faculty of Medicine and Pharmacy, University of Medicine, Pharmacy, Science and Technology “George Emil Palade” of Târgu-Mures, 540139 Târgu Mureș, Romania; benedekb.krisztina@gmail.com

**Keywords:** COVID-19, vascular inflammation, fat attenuation index, plaque vulnerability, thrombosis, chronic coronary syndrome, pericoronary adipose tissue

## Abstract

Inflammation is a key factor in the development of atherosclerosis, a disease characterized by the buildup of plaque in the arteries. COVID-19 infection is known to cause systemic inflammation, but its impact on local plaque vulnerability is unclear. Our study aimed to investigate the impact of COVID-19 infection on coronary artery disease (CAD) in patients who underwent computed tomography angiography (CCTA) for chest pain in the early stages after infection, using an AI-powered solution called CaRi-Heart^®^. The study included 158 patients (mean age was 61.63 ± 10.14 years) with angina and low to intermediate clinical likelihood of CAD, with 75 having a previous COVID-19 infection and 83 without infection. The results showed that patients who had a previous COVID-19 infection had higher levels of pericoronary inflammation than those who did not have a COVID-19 infection, suggesting that COVID-19 may increase the risk of coronary plaque destabilization. This study highlights the potential long-term impact of COVID-19 on cardiovascular health, and the importance of monitoring and managing cardiovascular risk factors in patients recovering from COVID-19 infection. The AI-powered CaRi-Heart^®^ technology may offer a non-invasive way to detect coronary artery inflammation and plaque instability in patients with COVID-19.

## 1. Introduction

### 1.1. The Impact of Vascular Inflammation in Atherosclerosis

It is well-established that vascular inflammation is a key factor in the development and progression of atherosclerosis. Plaques that are stable have a chronic inflammatory response, while those that are vulnerable or have ruptured have an “active” inflammation that can compromise the protective fibrous cap and raises the risk of rupture, which is the main cause of acute vascular events (AVE) [1].

Atherosclerosis is a prevalent condition that affects the blood vessels and increases the risk of serious AVE, such as acute coronary syndrome (ACS) and cerebrovascular events. These events occur when the plaques rupture and form clots, which can lead to severe complications and even death. The concept of “vulnerable plaque” (VP) was introduced to describe the unpredictable nature of the progression of atherosclerosis [2], a VP is one that is more likely to rupture and cause an AVE [3]. In recent years, research has focused on understanding the mechanisms that lead to plaque instability and rupture. However, there are also factors outside of the plaque that can increase a person’s risk for an event, leading to the concept of “vulnerable patient” [3,4]. A vulnerable patient is someone who is at a higher risk of developing an AVE or sudden cardiac death based on factors such as the characteristics of the plaque, perivascular inflammation, blood flow, and myocardial vulnerability [3,4,5].

In recent years, epicardial adipose tissue (EAT) and perivascular adipose tissue (PVAT) have been extensively researched for their roles as markers and promoters of local coronary inflammation [6,7]. PVAT is a layer of fat that surrounds the coronary arteries and other blood vessels, and is composed of lipid-filled cells (adipocytes), connective tissue cells (such as preadipocytes), and interstitial tissue. PVAT is considered part of the vessel and has a close anatomical and physiological relationship with the arterial wall [8]. EAT and PVAT also contribute to systemic inflammation by releasing cytokines into the circulation through paracrine signaling pathways [9]. Non-invasive imaging methods have been used to effectively quantify EAT and PVAT, which has been shown to predict the risk of AVE across a variety of cardiovascular disorders, and also correlates with markers for systemic inflammation, increased oxidative stress, severity of atherosclerosis, and patient prognosis [10,11].

### 1.2. COVID-19 Inflammatory Response: Pathophysiology

The COVID-19 pandemic has had a considerable impact on cardiovascular health, due to its effects on inflammation and destabilization of the immune system, both during the acute phase of the infection and over the course of the disease, as well as in long-COVID syndrome [12,13]. SARS-CoV-2 infection can exacerbate underlying cardiovascular diseases (CVD) or promote vulnerability of coronary atherosclerotic plaques due to high levels of pro-inflammatory cytokines and chemokines in the bloodstream. These substances can lead to microvascular and vascular thrombosis, coronary vasospasm, modulation of shear stress, and platelet activation, which can further contribute to plaque vulnerability [14]. The intensity of the cytokine storm, or excessive release of cytokines, has been identified as a significant component in predicting the clinical progression of extrapulmonary organ failure and mortality in COVID-19 patients [15].

### 1.3. PVAT-FAI Mapping for Inflammation Detection

The optimal method to detect inflammation in the PVAT using coronary computed tomography angiography (CCTA) is through the use of the fat attenuation index (FAI). The FAI is a measure of the CT attenuation gradients in the perivascular space and is calculated by comparing the attenuation values of the PVAT and the adjacent vessel. In cases of coronary inflammation, the composition of the PVAT undergoes a change towards higher aqueous and lower lipophilic content, which can be observed in the FAI due to a decrease in the CT attenuation of the PVAT. Chronic inflammation can also lead to unfavorable fibrotic and other remodeling of the PVAT, which can be identified using progressed radiomic texture analysis of the perivascular adipose tissue [16,17]. The FAI has several advantages as a measure of vascular inflammation; it is not affected by the degree of coronary calcification, is not influenced by individual systemic inflammation as measured by hs-CRP, and is not associated with the severity of coronary stenosis [18,19].

CCTA is commonly used in COVID-19 patients with diagnosed or suspected CVD, especially when the results are likely to significantly affect the patient’s treatment plan or save their life [8,20]. As previously discussed, local vascular inflammation can be quantified non-invasively using CT imaging with the computation of the FAI, using artificial intelligence algorithms [21,22].

## 2. Results

### 2.1. Baseline Characteristics of the Study Population

This study recruited 158 patients who reported chest pain and had a low to intermediate clinical likelihood of CAD. In total, 47.46% of the participants (*n* = 75) had a history of COVID-19 infection a few months prior to the CCTA examination, while the remaining 52.53% did not have any prior infection. The participants had an average age of 61.63 ± 10.14 years, and 67.08% (*n* = 106) of them were male. There were no statistically significant differences between the two groups in terms of gender, age, and comorbidities such as hypertension, diabetes, and obesity. In contrast, it was observed that group 1 exhibited a lower incidence of hypercholesterolemia in comparison to group 2 (40.00% vs. 59.04%, *p* = 0.02), alongside having a higher body mass index (28.51 ± 4.21 vs. 26.93 ± 4.25, *p* = 0.03) and significantly reduced levels of triglycerides (134.1 ± 75.04 vs. 154.0 ± 64.58, *p* = 0.03). The groups did not show any significant differences in terms of PCI after CCTA, multi-vessel PCI, heart failure, left ventricular ejection fraction (LVEF), serum creatinine, and total cholesterol. The average time from COVID-19 infection to CCTA examination for group 2 was 138.1 ± 103.2 days and there were no significant differences between the groups in terms of vaccination against COVID-19. The baseline characteristics, comorbidities, and risk factors of the two study groups are presented in Table 1.

Figure 1 demonstrates the case of a single patient who underwent CCTA among the 158 patients referred to the partner center for further assessment utilizing AI algorithms.

### 2.2. PVAT-FAI Values and Scores

According to the standard adipose tissue Hounsfield unit range (−190 to −30 HU), there were no significant differences in the coronary FAI index values between the two groups as revealed by the conventional CT scan. The FAI-score was consistently higher in the non COVID-19 group; more precisely: LAD (group 1—9.32 ± 6.00 vs. group 2—11.61 ± 7.60, *p* = 0.05), LCX (group 1—10.48 ± 6.24 vs. group 2—12.43 ± 6.65, *p* = 0.05), FAI-score TOTAL (group 1—10.47 ± 7.19 vs. group 2—12.81 ± 8.28, *p* = 0.001) (Table 2).

### 2.3. PVAT-FAI Score Centile of Coronary Inflammation

The FAI-score of the LAD, LCX, and RCA was used to create percentile curves across various age and sex groups. These curves were evaluated for their prognostic significance in Cox models that were adjusted for hypertension, diabetes mellitus, smoking, hyperlipidemia, high-risk plaque characteristics, and the modified Duke prognostic CAD index (Figure 2).

For the FAI-score Centile, the overall pattern shifts significantly, as the values for all three coronary arteries are higher for the subjects in the COVID-19 positive group, as follows: FAI-Score Centile LAD (group 1—0.66 ± 0.29 vs. group 2—0.58 ± 0.28, *p* = 0.05), FAI-Score Centile LCX (group 1—0.79 ± 0.16 vs. group 2—0.68 ± 0.26, *p* = 0.03), FAI-Score Centile RCA (group 1—0.83 ± 0.20 vs. group 2—0.68 ± 0.29, *p* = 0.05). FAI-Score Centile values of the two study groups are presented in Figure 3.

## 3. Discussion

A major challenge in the field of coronary atherosclerosis research is to find markers that can detect coronary inflammation, which is closely connected to the vulnerability of plaques. By doing so, it would be possible to identify patients who are at an elevated risk of AVE even before they show any clinical symptoms. Additionally, addressing inflammation may be a viable approach for preventing and managing this disease [23].

The ongoing coronavirus pandemic has had a significant impact on our society, including a significant effect on cardiovascular health [24], which was initially thought to be primarily a respiratory illness. The increased incidence of COVID-19 has presented healthcare professionals with various challenges, including assessing inflammation in PVAT as a marker of cardiovascular involvement. The respiratory issues caused by COVID-19 can lead to severe hypoxemia, which can result in multi-organ failure and cardiac damage [25]. The virus induces inflammation by triggering the release of cytokines and chemokines from respiratory epithelial cells, dendritic cells, and macrophages, as well as promoting the accumulation of EAT and PVAT. This can lead to local vascular inflammation and endothelial dysfunction, resulting in plaque formation and progression [26,27]. Certain biomarkers, such as inflammatory markers, chest CT findings, and nutritional status scores, have been shown in studies to predict adverse outcomes in COVID-19 patients. These results suggest that early identification and management of high-risk patients could lead to better outcomes [28,29,30].

Several studies have reported that COVID-19 infected patients have a higher incidence of cardiovascular events, including acute myocardial infarction, stroke, and heart failure, within one year of infection [31]. Additionally, Katsoularis et al. conducted a self-controlled case series and matched cohort study in Sweden, which showed an increased risk of acute myocardial infarction and ischemic stroke following COVID-19 [32]. Moreover, the Liuzzo and Volpe article highlighted the substantially elevated long-term cardiovascular risk linked to SARS-CoV-2 infection [33]. Collectively, these studies suggest that COVID-19 may have a significant impact on cardiovascular health and can potentially lead to long-term consequences. The importance of closely monitoring and managing cardiovascular health in COVID-19 patients is emphasized, and further research is necessary to fully comprehend the cardiovascular effects of COVID-19.

CCTA has become the first-line test for investigating suspected CAD. However, as its use expands in different clinical settings, it is essential to enhance its accuracy while also improving its diagnostic and prognostic importance in the early stages of coronary atherosclerosis [34,35,36]. The main findings of our study are as follows.

Firstly, our study revealed that the study population had a high FAI Score, and in all cases, the FAI Score was consistently higher in the non COVID-19 group. The new FAI Score for each coronary artery is a more dependable way of measuring coronary inflammation and a more accurate predictor of the risk of cardiac mortality than previous FAI methods [16,24].

Secondly, we discovered that the existence of lesions with greater pericoronary FAI-Score Centile value in the studied population were more commonly associated with patients who had previously been infected with COVID-19. It is established that the FAI-Score tends to increase with age. The percentile curves associated with the FAI Score are estimates of how the score is distributed among a group of patients who have undergone a clinically indicated CCTA. These curves provide a way to compare an individual’s FAI Score to the others in the same age and gender group. It is important to note that the FAI Score is just one factor among many that can affect an individual’s overall cardiovascular risk. In our opinion, this may be the reason why the COVID-19 group shows higher values of FAI-Score Centile.

Our study suggests that the PVAT-FAI Score is a trustworthy marker of coronary immune-inflammatory activation and its correlation with plaque vulnerability. This has important implications for assessing vascular involvement beyond typical conditions encountered by cardiologists, particularly in the context of the COVID-19 pandemic [37]. Further research is needed to understand the impact of plaque location on hemodynamic characteristics and the relationship between the specific vulnerable plaque phenotype and the degree of PVAT inflammation.

In the last decade, CCTA has been established as an excellent comprehensive tool for investigating suspected coronary artery disease. Mapping the PVAT-FAI on routine CCTA can detect coronary artery inflammation non-invasively by measuring changes in the composition of pericoronary fat. At the same time, systemic inflammation can be enhanced due to various infectious diseases, including SARS-CoV-2 infection. Thus, it is postulated that COVID-19 can modulate pericoronary FAI via systemic inflammatory pathways and plaque composition towards a higher vulnerability degree, which, in turn, can alter patient prognosis. There are scarce data regarding the evolution of FAI, and, therefore, of local coronary inflammation following SARS-CoV-2 infection, which could interfere with the development and progression of coronary atherosclerosis.

## 4. Materials and Methods

### 4.1. Study Design and Population

Our cross-sectional observational study, conducted at the Center of Advanced Research in Multimodality Cardiac Imaging, Cardio-Med Medical Center (Târgu Mureș, Romania) included 158 patients presenting with pain in the chest area and a low to intermediate likelihood of CAD. The patients underwent 128-slice CCTA to assess coronary anatomy and atherosclerosis, determine FAI score, and perform plaque analysis. The study participants were categorized into two main groups based on their COVID-19 status: group 1 (*n* = 75) included patients who had a previous COVID-19 infection and group 2 (*n* = 83) included patients matched for age and gender who did not have a COVID-19 infection prior to the CCTA examination. We ensured the accuracy of patient selection in the first group by only including individuals who had laboratory-confirmed COVID-19, exhibited mild to moderate symptoms, and had not received any treatment in healthcare facilities. Demographic and laboratory characteristics, cardiovascular risk factors, and symptom development were all monitored and assessed for each patient before the CCTA examination.

Patients with high coronary calcification, irregular heart rate, inability to achieve heart rate below 65 beats per minute, morbid obesity, inability to follow breath-hold instructions, or any other condition that could affect image quality were excluded from the study. In addition, those with chest pain caused by factors other than CAD, a history of myocardial infarction, high clinical probability of CAD, severe symptoms, or ACS requiring percutaneous coronary intervention (PCI) were also excluded.

### 4.2. CCTA Acquisition Procedure and Image Post-Processing

All participants underwent a CCTA scan using a 128-slice Siemens Somatom Definition AS (Siemens Healthcare, Erlangen, Germany) from the Centre of Cardio-Med (Târgu Mureș, Romania). The scan was retrospectively gated with a heart rate below 65 beats per minute and the parameters included a tube voltage of 120 kV, gantry rotation time of 0.33 s, and a collimation of 128 × 0.6. Beta-blockers were given to patients with a resting heart rate above 65 beats per minute, and blood pressure was monitored during the administration of intravenous or oral beta blockers. The acquisition process began with a native scan for coronary calcium assessment, followed by the administration of 80–100 mL of iodine-based contrast material based on body weight with a 50-mL saline chase at a flow rate of 5.5–6 mL/s during inspiratory breath-hold. All obtained scans were saved in a devoted electronic imaging database for offline image post-processing and for cloud delivery.

All the acquired images were converted to DICOM format, then anonymized and transferred to the partner center (Centre of CARISTO Diagnostics, Oxford, UK) via the cloud for post-processing. The inclusion criteria for the analysis included the presence of plaques with a stenosis effect of at least 50% and with at least one vulnerability marker present (low attenuation noncalcified plaque—LAP, positive remodeling—PR, spotty calcifications—SC and the napkin ring sign—NRS). The FAI and AI-based FAI scores were determined for each of the major coronary arteries in all patients.

The calculation of the PVAT-FAI employs AI-enhanced algorithms (CaRi-Heart^®^, CARISTO Diagnostics, Oxford, UK) that deliver precise and consistent attenuation measurements in concentric 1-mm 3D layers of perivascular adipose tissue surrounding the human arterial wall [16,38]. The PVAT-FAI calculation process involves multiple complex steps, including segmenting the heart and analyzing the perivascular space and adipose tissue using AI algorithms. The algorithms correct for various factors and technical aspects of the scan to differentiate PVAT-FAI from conventional CT attenuation and accurately interpret coronary inflammation [39]. In order to enhance comprehensibility, Table 3 illustrates the distinction between FAI (HU) and FAI-Score.

### 4.3. Statistical Analysis

After quantifying the PVAT-FAI for each coronary artery, all collected data were sent back to our center and stored in an electronic Microsoft Excel database (Microsoft Corporation, Redmond, WA, USA). Statistical analysis was performed using GraphPad Prism 9.5 software (GraphPad Software, Inc., San Diego, CA, USA). The study included a PCAT-FAI analysis of 474 coronary arteries: 158 on the left anterior descending artery (LAD), 158 on the circumflex artery (LCX), and 158 on the right coronary artery (RCA). In addition, the CaRi-Heart^®^ risk and the Duke score were also determined for each individual [37].

The data were analyzed comparatively between patients with and without previous SARS-CoV-2 infection. Nominal (categorical) variables were reported as integer values (percentages) and compared between groups using the Chi-square test (χ^2^) and its variables. Numeric data were expressed as mean ± standard deviation, and a Mann–Whitney or unpaired Student’s *t*-test was used. Pearson correlation analysis was conducted to determine correlations between the PVAT-FAI and other variables as appropriate. Results were considered statistically significant if the two-sided *p*-value was 0.05.

## 5. Study Limitations and Future Directions

The purpose of this study was to assess the impact of COVID-19 infection on CAD in patients who underwent CCTA examinations for chest pain in the early stages after the infection, using the new AI—powered CaRi-Heart*^®^* medical device which combines standardized FAI mapping with plaque metrics and clinical risk factors to provide personalized cardiovascular risk assessment.

Our study has a few limitations that should be acknowledged. Firstly, the patients were only recruited from a single center, which may not accurately represent the entire population. Additionally, the study did not include a follow-up period, nor the measurement of serum inflammatory biomarkers, which could provide more insight into the correlation between coronary inflammation and risk of acute events. Secondly, the study only included lesions with at least 50% stenosis, so it is unclear if FAI-Score can identify cases with a high risk of events before significant stenosis. In addition, non-ST elevation ACS patients, who may have an increased inflammatory burden, were not included in the study (according to the 2020 ESC guidelines that recommend CCTA as a substitute for invasive angiography in cases with low to intermediate CAD likelihood). Although these limitations exist, additional research is required to validate the clinical significance of the FAI-Score in managing coronary inflammation.

In order to expand on the results of this proof-of-concept study, future research plans include conducting a follow-up period to evaluate the outcome prediction capability of the FAI mapping technique and its correlation with systemic inflammation as measured by biomarker analysis. Additionally, to address the limitations of this study, our research team intends to explore the correlation between local vascular inflammation and hemodynamic differences in vessels by computing shear stress using CCTA imaging. In an effort to gain a comprehensive understanding of overall coronary vulnerability, future studies will incorporate computerized plaque analysis, which provides insight into intrinsic vulnerability, in conjunction with FAI.

## 6. Conclusions

Lesions with higher pericoronary FAI-Score Centile values were more commonly found in patients who had previously been infected with COVID-19. The pericoronary FAI-Score percentile curves illustrate the level of coronary inflammation and provide a uniform interpretation of perivascular FAI mapping on CCTA. Patients in the upper percentile curves have an increased risk of cardiovascular disease, regardless of the presence of traditional risk factors or established atherosclerotic changes.

An association has been observed between COVID-19 infection and an elevated risk of destabilizing coronary plaque, as indicated by higher levels of inflammation in the pericoronary adipose tissue. The use of pericoronary FAI as a marker for identifying vulnerable patients at high risk for cardiovascular events could have significant implications in the deployment of targeted prevention strategies. However, further validation of these findings through larger studies with more consistent data is necessary to establish the utility of FAI in clinical practice.

## Figures and Tables

**Figure 1 ijms-24-07398-f001:**
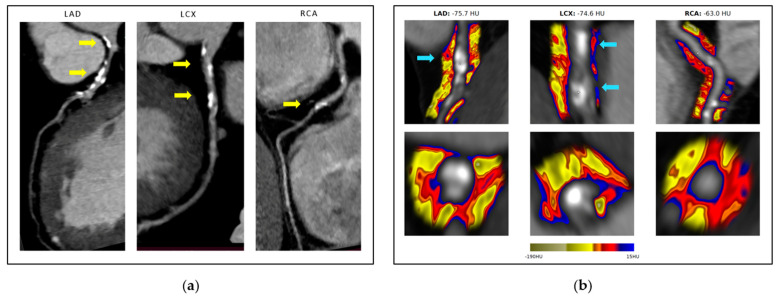
Conventional CCTA image of the three major coronary arteries (**a**) and a colored mapping representation of an abnormal FAI for the same patient (**b**). Figure shows CCTA images of the three major coronary arteries with a stable atherosclerotic lesion (yellow arrows) in a patient who had a COVID-19 infection a few months prior to CCTA examination (**a**) and delineated pericoronary fat with the FAI colored mapping around the non-culprit lesions (blue arrows) demonstrating abnormal FAI in the same patient (**b**). Using the CaRi-Heart^®^ v2.4.2 platform (panel **b**), FAI-Score was evaluated for each individual at baseline in the proximal LAD, LCX, and RCA.

**Figure 2 ijms-24-07398-f002:**
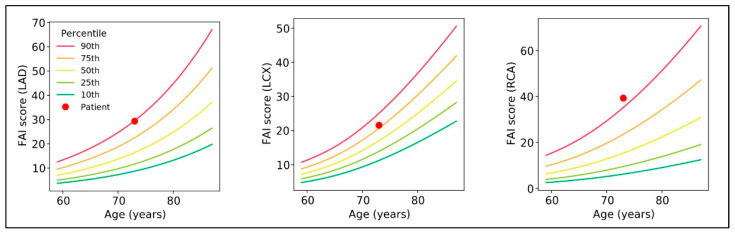
Nomograms with percentile curves for FAI-Score, adjusted for age, gender, and risk factors for each major coronary territory.

**Figure 3 ijms-24-07398-f003:**
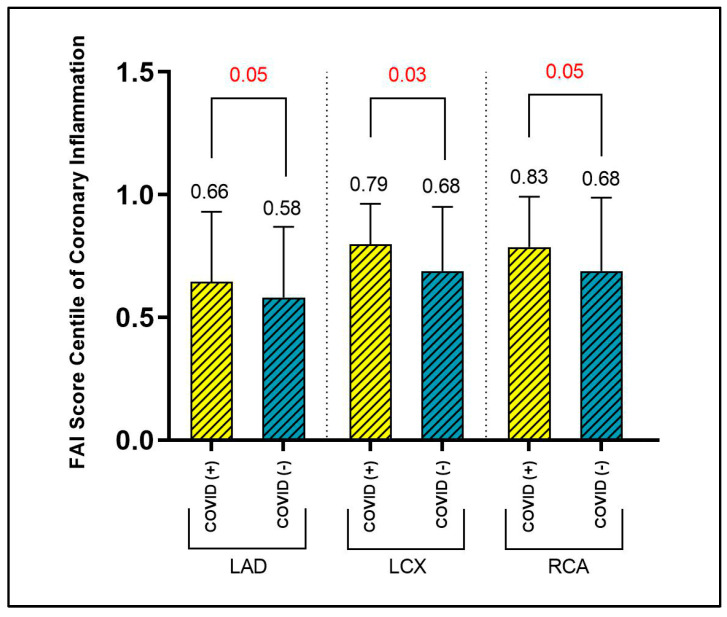
PVAT-FAI Score Centile of Coronary Inflammation for each coronary artery.

**Table 1 ijms-24-07398-t001:** Baseline characteristics, comorbidities, and risk factors in the study population.

Parameters	Whole Study Sample (n = 158)	Group 1 (COVID-19) (n = 75)	Group 2 (non COVID-19) (n = 83)	*p* Value *
Male gender, n (%)	106 (67.08%)	46 (61.33%)	60 (72.29%)	NS
Age at time of scan, mean ± SD	61.63 ± 10.14	60.29 ± 10.30	62.84 ± 9.90	NS
Smoking, n (%)	29 (18.35%)	10 (13.33%)	19 (22.89%)	NS
Hypertension, n (%)	135 (85.44%)	61 (81.33%)	74 (89.16%)	NS
Hypercholesterolemia, n (%)	79 (50.00%)	30 (40.00%)	49 (59.04%)	0.02
Diabetes, n (%)	44 (27.84%)	18 (24.00%)	26 (31.33%)	NS
Obesity, n (%)	41 (25.94%)	25 (33.33%)	16 (20.25%)	0.07
BMI, mean ± SD	27.57 ± 4.29	28.51 ± 4.21	26.93 ± 4.25	0.03
PCI after CCTA, n (%)	69 (43.67%)	27 (36.99%)	42 (50.60%)	NS
Multi-vessel PCI, n (%)	23 (14.55%)	7 (25.93%)	16 (38.10%)	NS
Heart failure, n (%)	117 (74.05%)	57 (76.00%)	60 (75.95%)	NS
LVEF (%), mean ± SD	47.69 ± 5.07	48.34 ± 4.18	47.12 ± 5.71	NS
Creatinine (mg/dL), mean ± SD	0.97 ± 0.26	0.93 ± 0.23	1.00 ± 0.27	NS
Total cholesterol (mg/dL), mean ± SD	167.3 ± 47.13	161.4 ± 43.55	171.0 ± 49.24	0.07
Triglycerides (mg/dL), mean ± SD	145.7 ± 69.49	134.1 ± 75.04	154.0 ± 64.58	0.03
COVID-19 vaccine, n (%)	99 (62.65%)	43 (57.33%)	56 (60.22%)	NS
Time from COVID-19 to CCTA (days), mean ± SD		138.1 ± 103.2		

* BMI—body mass index > 30 kg/m^2^; PCI—percutaneous coronary intervention; LVEF—left ventricular ejection fraction; NS—non-significance.

**Table 2 ijms-24-07398-t002:** PVAT-FAI Score Centile of Coronary Inflammation for Age and Gender.

Parameters	Whole Study Sample (n = 158)	Group 1 (COVID-19) (n = 75)	Group 2 (non COVID-19) (n = 83)	*p* Value *
FAI HU LAD, mean ± SD	−76.08 ± 7.66	−75.07 ± 7.59	−76.46 ± 7.74	NS
FAI HU LCX, mean ± SD	−71.32 ± 7.50	−71.44 ± 7.88	−71.21 ± 7.16	NS
FAI HU RCA, mean ± SD	−73.11 ± 8.94	−72.97 ± 9.38	−73.23 ± 9.61	NS
FAI-Score LAD, mean ± SD	10.54 ± 6.97	9.32 ± 6.00	11.61 ± 7.60	0.05
FAI-Score LCX, mean ± SD	11.48 ± 6.50	10.48 ± 6.24	12.43 ± 6.65	0.05
FAI-Score RCA, mean ± SD	15.00 ± 11.71	14.54 ± 12.17	15.40 ± 11.36	NS
FAI-Score TOTAL, mean ± SD	11.72 ± 7.87	10.47 ± 7.19	12.81 ± 8.28	0.001
FAI-Score Centile LAD, mean ± SD	0.61 ± 0.28	0.66 ± 0.29	0.58 ± 0.28	0.05
FAI-Score Centile LCX, mean ± SD	0.73 ± 0.22	0.79 ± 0.16	0.68 ± 0.26	0.03
FAI-Score Centile RCA, mean ± SD	0.73 ± 0.26	0.83 ± 0.20	0.68 ± 0.29	0.05

* By Mann–Whitney test or chi-square test or Fisher-exact test, when appropriate; Passed D’Agostino and Pearson normality test; NS — non-significance.

**Table 3 ijms-24-07398-t003:** The interpretation of FAI (HU) and FAI-Score.

Fat Attenuation Index (HU)	A non-adjusted, graphic illustration of the level of inflammation in the three primary epicardial coronary arteries.
Fat Attenuation Index-Score	A personalized measurement of the quantification of coronary inflammation in the three primary epicardial coronary arteries, adjusted for age and gender, expressed as a relative risk.

## Data Availability

The data presented in this study are available on request from the corresponding author. The data are not publicly available due to privacy reasons.

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
