# Peer review of "Elevated FAI Index of Pericoronary Inflammation on Coronary CT Identifies Increased Risk of Coronary Plaque Vulnerability after COVID-19 Infection"

_ijms, 2023, doi:10.3390/ijms24087398_

Round 1

Reviewer 1 Report

In the manuscript entitled “Elevated FAI Index of Pericoronary Inflammation on Coronary CT Identifies Increased Risk of Coronary Plaque Vulnerability After COVID-19 Infection”, Botond Mátyás et al presented the results of a cross-sectional observational study with interesting results regarding COVID-19 cardiovascular affectation. This study provides one of the potential explanations for the already reported increased cardiovascular risk in the patients who survived to the acute phase of COVID-19, and the results of the follow-up period to evaluate the outcome prediction capability of the FAI mapping technique will be quite interesting to see.

The conclusions of the manuscript are clear, and I think the content in general is adequate. This reviewer has only the following comments.

1.     The authors mentioned that group 1 (COVID-19 group) had a higher rate of hypercholesterolemia compared to group 2 (59.04% vs. 40.00%, p = 0.02). However, the data showed in the table 1 are inverse to what was expressed. 

2.     The authors did not mention the severity of COVID-19 presented in the patients of the group 1. Previous studies showed that the severity of the disease during the acute phase was related to worse outcomes (including higher rates of cardiovascular events) at long-term follow-up. In case they have access to these data it would be interesting to know. 

3.     I think that the potential relationship of these findings with the already reported worse cardiovascular outcomes in COVID-19 survivors compared with people without previous COVID-19 should be mentioned and discussed more extensively. Previous studies that address mid-term and long-term cardiovascular outcomes could be cited. Some of these studies are:

-       Ortega-Paz L, Arévalos V, Fernández-Rodríguez D, Jiménez-Díaz V, Bañeras J, Campo G, Rodríguez-Santamarta M, Díaz JF, Scardino C, Gómez-Álvarez Z, Pernigotti A, Alfonso F, Amat-Santos IJ, Silvestro A, Rampa L, de la Torre Hernández JM, Bastidas G, Gómez-Lara J, Bikdeli B, García-García HM, Angiolillo DJ, Rodés-Cabau J, Sabaté M, Brugaletta S; CV COVID-19 registry investigators. One-year cardiovascular outcomes after coronavirus disease 2019: The cardiovascular COVID-19 registry. PLoS One. 2022 Dec 30;17(12):e0279333. doi: 10.1371/journal.pone.0279333. PMID: 36583998; PMCID: PMC9803130.

-       Katsoularis I, Fonseca-Rodríguez O, Farrington P, Lindmark K, Fors Connolly AM. Risk of acute myocardial infarction and ischaemic stroke following COVID-19 in Sweden: a self-controlled case series and matched cohort study. Lancet. 2021 Aug 14;398(10300):599-607. doi: 10.1016/S0140-6736(21)00896-5. Epub 2021 Jul 29. PMID: 34332652; PMCID: PMC8321431.

-       Xie Y, Xu E, Bowe B, Al-Aly Z. Long-term cardiovascular outcomes of COVID-19. Nat Med. 2022 Mar;28(3):583-590. doi: 10.1038/s41591-022-01689-3. Epub 2022 Feb 7. PMID: 35132265; PMCID: PMC8938267.

Author Response

We extend our sincere gratitude for taking the time to provide your valuable feedback on our recent study. We deeply appreciate your positive remarks, and we are glad to hear that our discoveries were deemed significant and relevant. We are delighted that our cross-sectional observational study provided valuable insights into the association between the FAI mapping technique and COVID-19. We are heartened to learn that our research has contributed to furthering the understanding of this complex matter, and we are confident that it will serve as a valuable resource for fellow professionals in the field.

Reviewer 2 Report

This paper describes the association between COVID-19 and inflammation. It also discusses the usefulness of AI system of fat attenuation index (FAI).

The inflammatory relevance of COVID-19 is interesting. I don't have detailed knowledge about epicardial adipose tissue, difficult to understand the difference between FAI HU and FAI score, also how to calculate those. I would like to see a detailed explanation added.

Author Response

We would like to express our utmost appreciation for the time you took to provide us with your valuable feedback on our recent study. Your positive feedback is greatly appreciated, and we are pleased that our findings were considered to be significant and relevant. Our cross-sectional observational study has provided valuable insights into the correlation between the FAI mapping technique and COVID-19, and we are delighted that our research has contributed to a better understanding of this complex matter. We believe that our study will serve as a valuable resource for other professionals in the field and further enhance their knowledge in this area.
